# Structured Early Follow-Up in Head and Neck Squamous Cell Carcinomas: A Retrospective Cohort Study

**DOI:** 10.3390/biomedicines13051246

**Published:** 2025-05-20

**Authors:** Philipp Dittmann, Bernhard Lehnert, Friedrich Ihler, Chia-Jung Busch, Markus Blaurock

**Affiliations:** 1Klinik für Mund- Kiefer- Gesichtschirurgie, Plastische Operationen, Universitätsmedizin Greifswald, 17489 Greifswald, Germany; philipp.dittmann@med.uni-greifswald.de; 2Klinik für Hals-, Nasen-, Ohrenkrankheiten, Kopf- und Halschirurgie, Universitätsmedizin Greifswald, 17489 Greifswald, Germany; bernhard.lehnert@med.uni-greifswald.de (B.L.);

**Keywords:** HNSCC, follow-up, survivorship, staging

## Abstract

**Background/Objectives:** The various head and neck squamous cell carcinoma (HNSCC) subtypes are among the most common cancers globally, with significant recurrence rates within the first two years post-treatment. Despite advancements in treatment, structured early follow-up remains crucial for timely diagnosis and effective salvage treatment. **Methods**: This retrospective study examines the impact of implementing a structured initial restaging between three and six months after the conclusion of initial treatment. The study population included 532 patients treated with curative intent at the University Medicine of Greifswald, Germany, between 2010 and 2019. Patients were divided into two groups: standard follow-up (SF) and adapted follow-up (AF). The AF group received standardized post-treatment restaging, including imaging and panendoscopy or PET-CT exams. **Results**: We found a trend towards earlier diagnosis and a reduction in recurrences, although these differences were not statistically significant. Secondary cancers were observed more frequently in the AF group, significantly affecting overall survival. **Conclusions**: Our cohort supports structured initial cancer follow-up in HNSCC. Although not significant, an initial multimodal exam after treatment was well tolerated and showed a trend toward earlier diagnosis.

## 1. Introduction

The various types of head and neck squamous cell carcinoma (HNSCC) represent the sixth to eighth most common cancer worldwide [1,2,3] and the eighth to tenth most common cancer in Germany [4]. Most cases of head and neck cancer in Germany are associated with exposure to carcinogens such as smoking and alcohol consumption [5], but there is a growing number of oropharyngeal squamous cell carcinoma cases associated with oncogenic human papilloma virus subtypes [6,7,8,9]. Occupational hazards such as asbestos and polycyclic aromatic hydrocarbons make up most of the remaining cases [10,11].

There is currently no established screening program for the early detection of head and neck cancer in the general population [12]. Despite advancements in the treatment of head and neck cancers, including minimally invasive operative techniques and particle-based radiation therapy, the rate of recurrence and residual disease remains high [13,14]. The recurrence rate is particularly high in the first two years after the end of primary treatment [13,14]. Structured disease follow-up is essential for the early diagnosis of recurrences and allows effective salvage treatment. The interval between the diagnosis of the primary tumor and the recurrence significantly affects the patient’s prognosis, making early diagnosis crucial [15]. Despite this, there are no uniform standards in early follow-up of head and neck tumors [16]. The only consensus in the guidelines is the implementation of regular early follow-up examinations [17,18,19,20,21]. Early diagnosis is of special significance for immunotherapy, which significantly improved outcomes in palliative treatment but should begin before patients are symptomatic, as treatment effect is typically delayed. A large retrospective study has shown that 39% of HNSCC patients are asymptomatic at the time of recurrence diagnosis, making structured early follow-up exams essential [17]. This must be balanced with the financial impact of travel expenses, particularly in rural areas [22].

In our study, we examined the impact of implementing a structured initial restaging between three and six months after the conclusion of initial treatment.

## 2. Materials and Methods

### 2.1. Study Population

Our retrospective study is based on a population of patients with a first diagnosis of HNSCC who were treated with curative intent between 2010 and 2019 in the ENT department of the University of Medicine of Greifswald, Germany. Of the 532 patients included in the study, 501 (94.17%) were men and 31 (5.83%) were women. Only patients with histologically confirmed head and neck squamous cell carcinoma (HNSCC) of the oral cavity, oropharynx, larynx, and hypopharynx capable of curative treatment were included. Exclusion criteria were a noncurative treatment intention, patients who could not receive a full curative treatment dose due to comorbidities or patient preference, and persistent disease appearing before three months after the end of treatment. Furthermore, we excluded HNSCC of other locations and salivary gland cancer.

#### 2.1.1. Intervention

Before 2017, patients were seen for outpatient visits, including a clinical interview and examination, flexible endoscopy, and sonography every three months for the first two years. Further imaging and/or panendoscopy was performed on demand.

Starting in 2017, we offered patients at least one standardized post-treatment restaging at 3–6 months after the conclusion of cancer treatment. This included CT and/or MRI imaging and panendoscopy unless the primary site could be fully examined and was completely unremarkable. Patients following radiation with measurable nodal disease and hard-to-examine laryngeal cancer were offered PET imaging in accordance with the German insurance reimbursement.

Groups were therefore divided into the SF = standard follow-up = control group and AF = adapted follow-up = intervention group.

Patient interviews included questions on general well-being, weight loss, fever, night sweats, dysphagia, bleeding or other side effects, and persistent nicotine and alcohol use.

Case files were stratified by gender, age, primary tumor location, classification according to TNM classification of the Union for International Cancer Control (UICC), tumor stage, primary therapy performed, completion of primary therapy, time of recurrence (determined from initial diagnosis), recurrence location, recurrence therapy, recurrence-free interval, and the method of disease follow-up. A follow-up interval of at least five years was observed for all patients if possible. Furthermore, nicotine/alcohol abuse (nicotine in pack years, alcohol abuse above the safe amount for women of 12 g pure ethanol and men of over 24 g pure ethanol per day as defined by the Federal Center for Health Education in Germany (BZgA—Bundeszentrale für Gesundheitliche Aufklärung)).

Study planning and reporting was structured according to STROBE criteria for cohort studies where applicable [23].

#### 2.1.2. Statistics and Data Analysis

Data analysis and graphs were created in R 4.4.3 (R Foundation for Statistical Computing, Vienna, Austria) and GraphPad Prism 10 (GraphPad Software, San Diego, CA, USA). We used R [24] with the additional packages ggplot2 [25], survival [26], survminer [27,28], forestmodel [29] and gtsummary [30].

## 3. Results

We included a total of 532 patients in the study, comprising 401 patients in the control group (SF) and 131 in the intervention group (AF). Specifics for both groups are found in Table 1. An additional 125 patients were excluded during screening for the following reasons: uncommon tumors such as salivary gland and nasopharyngeal cancers, palliative intent, premature treatment discontinuation, metastatic disease, death before follow-up and loss to follow-up.

There was a difference in age distributions (*p* = 0.023), the AF being slightly older. Age in decades is compared to better illustrate group differences. Tumor extension is compared by TNM, though we did not evaluate affixes for simplification. T and N are included; M is not listed, as patients with distant metastases were excluded. UICC Tumor stages were stratified, stage 4 being the most common in both groups. There were more stage 1 cases in the AF group (18% SF and 29% AF) and fewer stage 4 cases (49% SF and 37% AF). Alcohol and nicotine abuse was comparable in the SF and AF groups, with a slight reduction in the AF group. Secondary cancers were observed about twice as often in the AF group (SF 13% vs. AF 27%). Despite the AF group being more recent, the average follow-up time was longer (SF 207 weeks (90, 316); AF 263 weeks (156, 304)). This was even more pronounced in the patients experiencing recurrence (SF 118 weeks (76, 222); AF 213 weeks (81, 293)), though the smaller numbers meant the result was not statistically significant. Treatment types were shifted towards lower intensity treatment in the AF group compared to SF, with more surgery alone (44% to 29%) and less surgery and chemoradiation (19% to 29%). Recurrences were more often treated with curative intent in the AF group (57%) than the SF group (34%). Seven patients with oro- or hypopharyngeal cancer and 16 patients with oral cavity cancer were HPV positive in the AF cohort. Of these patients, 13 were active smokers. Inclusion or exclusion did not significantly change results and was therefore not listed separately. Testing in the AF was not performed reliably, and not a single HPV-positive case was found in our series.

Overall mortality does not differ significantly, as seen in Figure 1. There is a clear negative correlation between tumor stage and mortality. A higher age was also associated with an increased risk of death, as were nicotine and alcohol consumption. Secondary cancers conferred the highest risk of death with a hazard ratio of 2.09 (1.53–2.06) in both the SF group and the AF groups.

Survival did not significantly differ between groups, as shown in Figure 2. Cause of death could not be established with enough certainty in many cases to differentiate between deaths caused by the HNSCC and deaths caused by secondary cancers and other diseases.

Figure 3 shows the recurrence-free survival for both groups. There is a trend toward earlier diagnosis and an overall reduction in recurrences. Neither difference is statistically significant.

Figure 4 more closely describes the recurrence patients in the AF cohort, including the location of the recurrence and the exam that led to diagnosis. Most recurrences were identified by examination, followed by imaging and panendoscopy. Of note, only very few patients (9%) in that cohort were diagnosed after self-presentation, while the vast majority of cases were found during follow-up exams.

The bar chart in Figure 5 shows the number of recurrences in both cohorts and their localization. As subsites are listed separately, single patients might have multiple events in this visualization. Figure 6 shows the number of recurrences per year by patient compared to healthy patients remaining in follow-up for both groups combined. The percentages of patients with recurrences are 18.6% (1 in 5) in the first year, 10.4% (1 in 10) in the second year, 3.4% (1 in 30) in the third year, 4.1% (1 in 24) in the fourth year and 4.5% (1 in 22) in the fifth year.

## 4. Discussion

Introducing a standardized initial staging after the conclusion of cancer treatment was well received by our patients, as demonstrated by the high participation of 86% in an elective exam. Unfortunately, our initial hypothesis of significantly decreasing time to diagnosis for recurrences could not be shown in our cohort, though we see a trend evident when observing the graph in Figure 3. The intervention coincided with certification with the German Cancer Society (DKG—Deutsche Krebsgesellschaft) and was one of several measures in the standardization of treatment processes and procedures. This explains why follow-up time in the intervention group was significantly longer despite being the more recent cohort. Additionally, access to most of the death records was available through registry information starting in the intervention cohort, further skewing results.

The longer observation found significantly more secondary cancer in the intervention group. Secondary cancer was the highest predictor of mortality that matched similar observations through the SEER database [31,32]. It is likely that this is not due to any change in secondary cancer rates but another sign of a more stringent observation of the intervention group. This also explains why mortality was similar between groups despite there being fewer recurrences, and these more likely having curative treatment strategies.

Unfortunately, we could not generate any significant evidence on HPV-related disease, as we had low rates in the AF group and no reliable data in the SF group, with not a single positive test, likely due to insufficient diagnostics. It is of note that more than half of the patients were smokers (13/23), which might contribute to their group not being significantly different than the HPV-negative patients.

Most head and neck squamous cell carcinomas (HNSCC) exhibit recurrences within the first two years post-treatment, and follow-up examinations are rightfully focused during this timeframe [33,34]. The NCCN guidelines stipulate that patients be examined every one to three months during the first year, with a reduction to every six to twelve months thereafter. The German guidelines, such as the most recent version for oro- and hypopharyngeal cancer, suggest visits every three months for the first two years and every half year until five years have passed [35]. Use of guidelines in this timeframe is part of the DKG certification process that has been shown to improve outcomes on a population level in Germany [36].

Imaging is of most use to patients at higher risk of recurrence and is of most value within the first six months post-treatment [37]. While regular imaging within the first six months post-treatment is recommended, studies suggest that beyond this period, imaging should be reserved for cases with suspicious clinical findings, emphasizing the need for evidence-based guidelines to inform clinical decisions in follow-up care [38,39]. Patients with extranodal extension and advanced disease have the highest risk for distant metastases that are mostly found in the lung [40]. This suggests that these patients are most likely to benefit from additional imaging toward the end of the first year after treatment, possibly via PET-CT [41].

When choosing an imaging modality, there is strong evidence for the use of PET-CT in follow-up. The high negative predictive value of around 95% is especially useful to rule out active disease in residual nodal findings after radiation therapy [42]. Unfortunately, the high negative predictive value is contrasted by the low positive predictive value of 58.6 for primary sites and 52.1 for the neck [41]. This requires the correlation of the findings with other modalities.

Neck ultrasound, especially when using advanced algorithms, can significantly improve the positive predictive value to 87% when differentiating neck nodes [43].

Clinical exams, including ultrasound and endoscopy, remain the mainstay of follow-up exams, as suggested in the German guidelines [35]. Advanced endoscopy and modalities such as NBI (narrow band imaging) might further improve detection of early recurrences, especially in the postoperative setting [44]. The role of panendoscopy remains unclear, as previously found by Muenscher et al. [45]. That study was somewhat marred by the late onset of panendoscopy after one year, when a large part of recurrences had already been diagnosed.

For our practice, we found the following conclusion from the data for early follow-up:-An initial exam should be performed 3 months after completion of treatment, including evaluation of symptoms, flexible videoendoscopy, and ultrasound. We strongly suggest panendoscopy if there is any uncertainty in evaluating the primary site.-Stage I-II disease should receive at least one initial imaging modality, such as magnetic resonance imaging or cervical CT, although T1a laryngeal cancer might not need additional imaging.-Stage III and above should receive imaging at 3–6 months that includes the thorax and upper abdomen to detect lung and liver metastases and be offered additional imaging at least within the first two years.-Stage IV disease should receive imaging at 3–6 months that includes the thorax and upper abdomen to detect lung and liver metastases, and it is strongly suggested that at least one additional imaging be provided towards the one-year mark to detect delayed asymptomatic distant metastases. Later imaging should be considered depending on comorbidities and patient compliance.

We find little conclusive evidence from our data to suggest the use of standardized screening for distant metastases besides the thorax and upper abdomen. Besides skin involvement, we only found three patients with bone metastases and two patients with liver metastases. Three of the four had multiple metastases in other areas as well. All but one were covered by imaging the neck and thorax to the upper abdomen.

After the initial follow-up phase, the evidence on the frequency of exams is much less clear. The low likelihood of detection must be weighed against the financial toxicity of repeated visits. This is especially true for more rural, low-income areas. Our area includes islands that often necessitate a drive of several hours to clinics and additional travel for outpatient CT or MRI exams.

Measures such as patient-reported outcome measures might play a significantly more important role in this phase of follow-up [46]. This must be weighed against the low rate of self-presentation we found in our data, as seen in Figure 4. Further research into ways of better recognizing high-risk diseases might focus resources on these patients. This could be achieved through more advanced pathological risk stratification, such as the use of miRNA-based subtyping [47], but more significantly by including ctDNA-based surveillance. Studies such as LIONESS have successfully used ctDNA as a means of predicting recurrence even late during treatment, though reimbursement for these methods depends on larger studies in the future [48].

All the aforementioned substratification does not help with the second primary disease. Our data finds multiple late recurrences with aggressive presentation that were classified as recurrences, though their presentation makes a second primary likely. Previous data suggests that many of these might be second primary tumors [49]. This is also relevant for distant metastases that might be second primary cancers in up to a third of cases [50].

## 5. Conclusions

Our cohort confirms the focus on structured initial cancer follow-up in HNSCC. Though we do not achieve significance, we find that an initial multimodal exam after treatment conclusion was well tolerated and showed a trend toward earlier diagnosis.

Patterns of further recurrences suggest risk-stratified imaging in the latter half of the first year and the second year in high-risk cases.

We confirm previous publications that find a high rate of secondary cancers significantly affecting OS in this patient collective. Late follow-up after more than two years remains a challenging topic that might be improved by ctDNA and other methods of risk stratification in the future, though second primaries might be more common than currently thought.

## Figures and Tables

**Figure 1 biomedicines-13-01246-f001:**
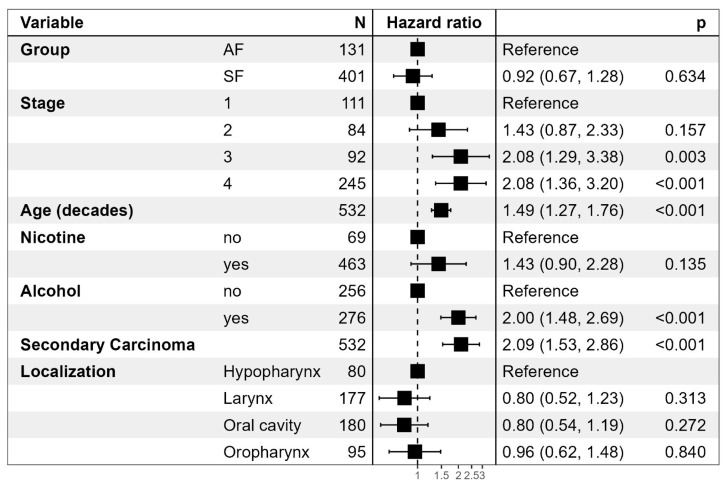
Forest plot comparing subgroups based on different characteristics, including hazard ratios, standard deviations and *p*-values. HPV subgroups were unreliable in the SF group and not significant in the AF group and therefore not included separately.

**Figure 2 biomedicines-13-01246-f002:**
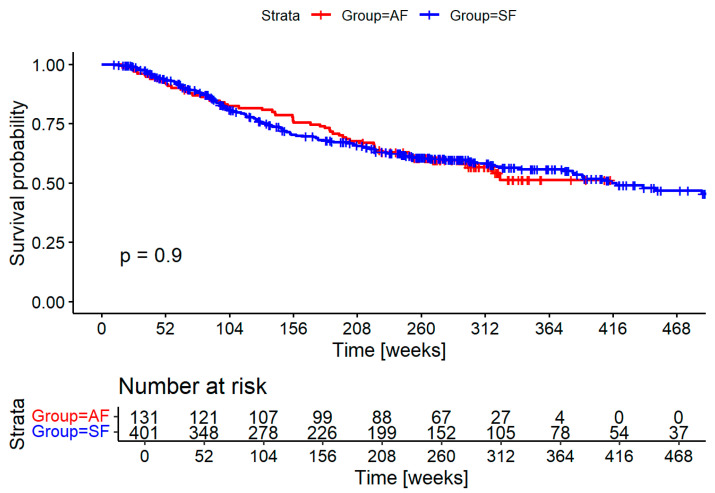
Survival probability of the adapted follow-up (AF in blue) and standard follow-up (SF in red) groups in weeks. Numbers at risk are shown for yearly increments.

**Figure 3 biomedicines-13-01246-f003:**
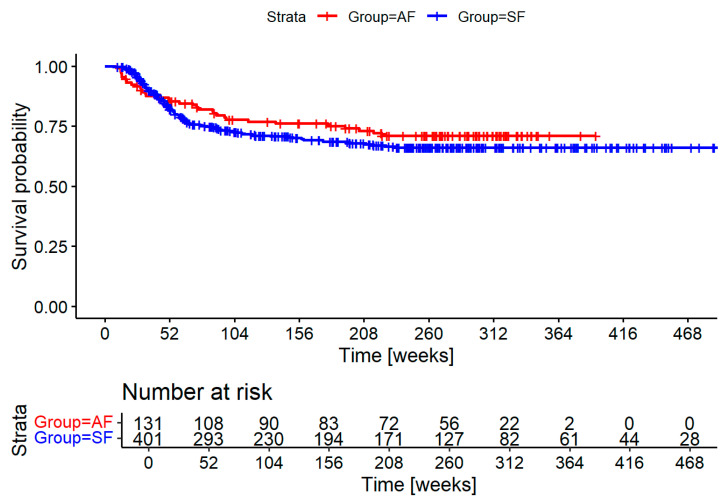
Recurrence-free survival of the adapted follow-up (AF in red) and standard follow-up (SF in blue) groups in weeks. Numbers at risk are shown for yearly increments.

**Figure 4 biomedicines-13-01246-f004:**
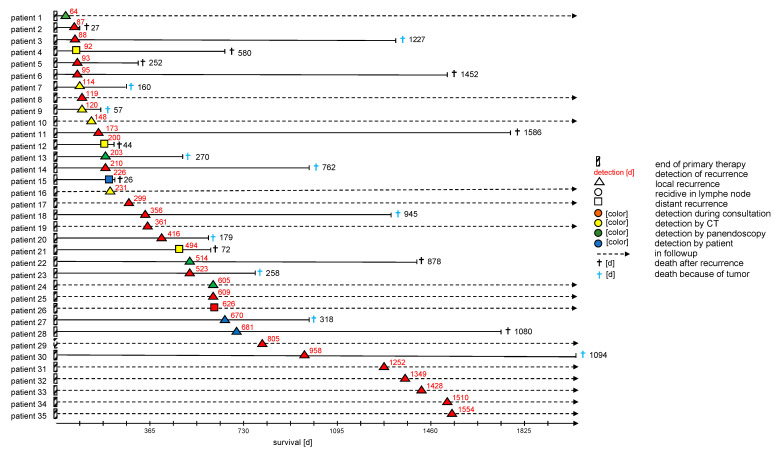
The figure describes all 35 recurrence patients from the AF group. It shows the duration until the diagnosis of recurrence from the end of primary therapy in days (red numbers), the primary location of the recurrence as indicated by the shape, how the recurrence was diagnosed by color and how many days the patient survived after diagnosing the recurrence (black number or arrow if still alive). Death is stratified by either being related to the recurrence (black cross) or for other causes not directly linked (black cross).

**Figure 5 biomedicines-13-01246-f005:**
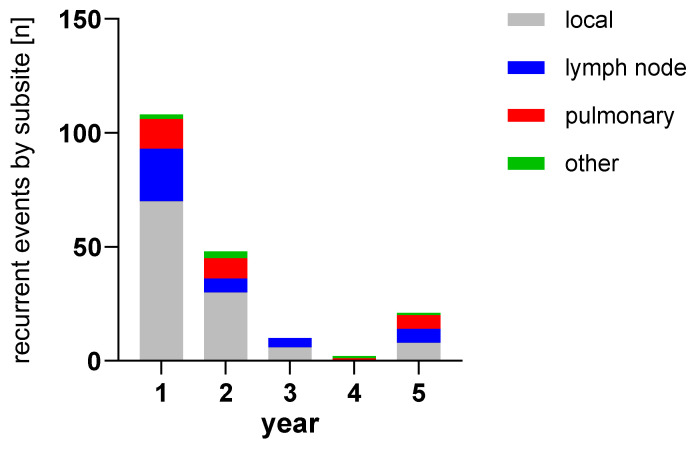
Recurrence findings by subsite event per year. Single patients might display findings in multiple subsites, so the results do not correspond to absolute patient numbers. Colors indicate the location of findings.

**Figure 6 biomedicines-13-01246-f006:**
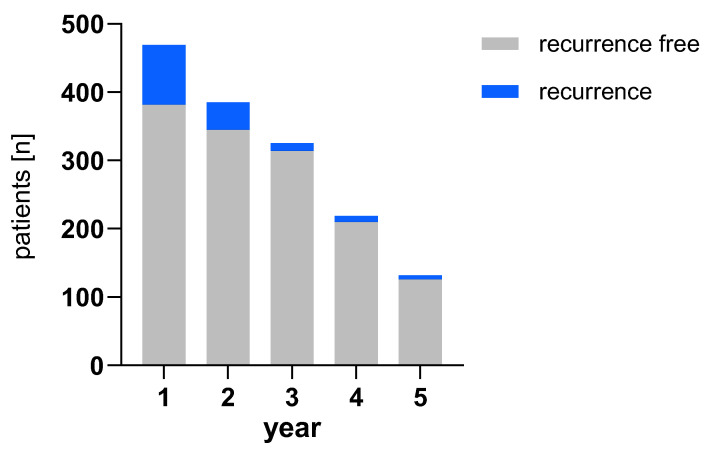
Patients with recurrence (in blue) compared to patients who remain alive in follow-up (in gray) per year.

**Table 1 biomedicines-13-01246-t001:** Comparison of patient characteristics between the standard follow-up (SF) and adapted follow-up (AF) groups. Second primary includes all cancer entities. HPV subgroups were unreliable in the SF group and not significant in the AF group and, therefore, not included separately.

Characteristics	SF n = 401 ^1^	AF n = 132 ^1^	*p*-Value ^2^
Age	59 (53, 67)	61 (56, 68)	0.023
T stage			0.13
1	106 (26%)	47 (36%)	
2	120 (30%)	37 (28%)	
3	74 (18%)	25 (19%)	
4	101 (25%)	23 (17%)	
N stage			0.031
0	184 (46%)	70 (53%)	
1	54 (13%)	19 (14%)	
2	151 (38%)	34 (26%)	
3	12 (3.0%)	9 (6.8%)	
UICC Stage			0.026
1	73 (18%)	38 (29%)	
2	65 (16%)	19 (14%)	
3	66 (16%)	26 (20%)	
4	197 (49%)	49 (37%)	
Followup Exam			<0.001
Yes	156 (39%)	113 (86%)	
No	245 (61%)	19 (14%)	
Alcohol			<0.001
Yes	225 (56%)	51 (39%)	
No	176 (44%)	81 (61%)	
Nicotine			<0.001
Yes	367 (92%)	97 (73%)	
No	34 (8.5%)	35 (27%)	
Second Primary	54 (13%)	35 (27%)	<0.001
Follow-Up Time (weeks)	207 (90, 316)	263 (156, 304)	0.090
Localization			0.10
Hypopharynx	62 (15%)	18 (14%)	
Larynx	131 (33%)	47 (36%)	
Oral cavity	128 (32%)	52 (39%)	
Oropharynx	80 (20%)	15 (11%)	
Treatment Type			0.01
Surgery alone	119 (29%)	58 (44%)	
Surgery and chemoradiation	118 (29%)	25 (19%)	
Surgery and radiation	63 (15%)	22 (17%)	
Definitive chemoradiation	80 (20%)	18 (14%)	
Radiotherapy	21 (5%)	9 (7%)	
Recurrence treatment			0.02
Curative	39 (34%)	20 (57%)	
Palliative	76 (66%)	15 (43%)	

^1^ median (Q1, Q3); n (%). ^2^ Wilcoxon rank sum test; Pearson’s chi-squared test.

## Data Availability

Primary data are not available to protect patient anonymity.

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
