# Peer review of "Structured Early Follow-Up in Head and Neck Squamous Cell Carcinomas: A Retrospective Cohort Study"

_biomedicines, 2025, doi:10.3390/biomedicines13051246_

Round 1
Reviewer 1 Report
Comments and Suggestions for Authors
The study was conducted with proper methodology and is well-structured. Although statistical significance was not reached, the main finding—regarding the impact of implementing a structured initial staging between three and six months after the conclusion of the initial treatment—still showed a trend toward earlier diagnosis.
Minor review comments
• Please specify the nature of the study in the title (e.g., whether it is an RCT or a retrospective study).
• Indicate whether any reporting guidelines were followed in the manuscript preparation (e.g., CONSORT, STROBE, MOOSE, STREGA, etc.).
• Provide a more detailed description for the figures included.
Author Response
Thank you for the constructive criticism. Please find the following point-by-point commentaries:
Please specify the nature of the study in the title (e.g., whether it is an RCT or a retrospective study).
Thank you for the comment. We have changed the name accordingly.
Indicate whether any reporting guidelines were followed in the manuscript preparation (e.g., CONSORT, STROBE, MOOSE, STREGA, etc.).
Our manuscript and study design follows STROBE guidelines and these have been referenced accordingly.
Provide a more detailed description for the figures included
We have changed the descriptions.
Thank you again for the review!
Reviewer 2 Report
Comments and Suggestions for Authors
Dear authors,
First, I would like to congratulate you on your retrospective study on the impact of implementing a structured initial restaging – between three and six months – after the conclusion of initial treatment of patients with head and neck squamous cell carcinoma (HNSCC). This is a very relevant topic, given that HNSCC is one of the most common cancers worldwide and presents high recurrence rates in the first two years post primary treatment.
I have just a few questions and suggestions that could further enhance your work (please see the attached PDF file).

Author Response
Thank you for the constructive criticism. Please find the following point-by-point commentaries:
1) Page 1, line 21: Please delete the extra letter “A” at the beginning of the paragraph (conclusion)
Has been changed.
2) Page 4, lines 119 – 122: What about the HPV status in SF group?
There were several tests but no positive cases. It is unclear if this is due to inadequate testing or low cases. Evidence from other entities seems to indicate that there is a very low HPV rate in our region. We have elaborated more on the numbers in the text.
3) Page 7, lines 151 – 153: Do you consider that the smaller sample size of the AF group may have contributed to this lack of significance?
Yes. Unfortunately the pandemic has altered timelines so much that the cohort would have to be post 2022 since there are several time delays in the 2020 and 2021 data. That might be worth a followup with data from the years 2022-2025.
4) Page 8, lines 161 – 163: Would this highlight the relevance of the AF follow-up strategy?
Very much so, we have briefly mentioned this in page 11 line 269-272 when discussing ctDNA vs. Patient reported outcomes.
5) Page 8, lines 172 – 174: What group? This information is not clear enough. Please, better clarify it in the text.
Thank you. We have made it more clear that this is both groups.
6) Page 9, figures 4 and 5: Please, choose images with higher resolution.
The images are vector graphics and I presume there was some compression issue with the review copy. It should not be problem in the final draft. I have changed the first of the two diagrams as there was a border that made it into the draft
7) Page 11, lines 247 – 257: Why didn't the authors summarize these "conclusions" and include them in the Conclusion of the study?
Good point, we have changed the conclusion accordingly and added it to the abstract.
Thank you for your constructive review!
Reviewer 3 Report
Comments and Suggestions for Authors
Interesting article on a paramount aspect of cancer care.
Standard controls are significant for oncologic patients, as well as for the psychological aspect: these patients are frequently refractory to punctual controls because they are afraid of relapse.
Standard control protocol with frequent clinical visits and new CT DNA exams may improve relapse early detection.
Author Response
Thank you for your commentary. We also feel strongly that ctDNA will provide a method to better stratify patients for late follow-up and concur that follow-up exams place a lot of stress on patients.
Reviewer 4 Report
Comments and Suggestions for Authors
I’d like to congratulate P. Dittmann et al. for a nicely written work “The effect of structured early follow-up in head and neck squamous cell carcinomas”. It is always important to review the strategies in patient treatment and follow-up. I think this work could be published after some editing.
I think it is better to list statistical tests in materials and methods, not only under tables or charts, and following this remark, I think it would be informative to list specific libraries of R that were used for statistics or graphs, if any. It also helps with the recognition of other people's work.
In the results, it would be easier to read with information of significance near the results, not only in the table but also in the text. How many patients in group SF were HPV-positive? You only write about the AF group in this aspect. Was there some difference? In Table 1, there are some missing p-values for treatment type and recurrence treatment, which probably indicate significant differences.
Have you observed specific differences in patients who had a recurrence after two years? In Figure 3, there are those 7 patients with late recurrence, 29-35 patients, especially 31-35. I know it is a small group, but maybe there was something specifically different?
The discussion is nicely written, and I have only a minor comment. I think it would be informative which RNA-based subtyping could be used for risk stratification – I mean, kind of RNA like miRNA, like you’ve written ctDNA.
Author Response
Dear Reviewer,
Thank you for the kind and well written review. We have tried to implement the changes you suggested.
I think it is better to list statistical tests in materials and methods, not only under tables or charts, and following this remark, I think it would be informative to list specific libraries of R that were used for statistics or graphs, if any. It also helps with the recognition of other people's work.
We have included the libraries. The statistical tests you mention are meant to explain the method used in the table and we feel it is sensible to leave these in place. The problem is more that the table breaks the page barrier and we felt it better to leave the standard formatting and hope that the final typesetting gets everything on one page.
In the results, it would be easier to read with information of significance near the results, not only in the table but also in the text. How many patients in group SF were HPV-positive? You only write about the AF group in this aspect. Was there some difference? In Table 1, there are some missing p-values for treatment type and recurrence treatment, which probably indicate significant differences.
We have added the p-values and elaborated significantly on the HPV group. We have tried very hard to restructure the results by felt things get very confusing when we tried to restate more from the tables.
Have you observed specific differences in patients who had a recurrence after two years? In Figure 3, there are those 7 patients with late recurrence, 29-35 patients, especially 31-35. I know it is a small group, but maybe there was something specifically different?
I’m afraid that it is too speculative to really write in the article but anecdotally we find more "local recurrences" in the very late group which fits with the observation of Brakenhoff et al. that found second primary cancer when examining driver mutations of late “recurrences”.
The discussion is nicely written, and I have only a minor comment. I think it would be informative which RNA-based subtyping could be used for risk stratification – I mean, kind of RNA like miRNA, like you’ve written ctDNA.
We have made it more clear that we were referring to miRNA in that case.
Thank you!